# Investigation of Physicochemical Properties of the Structurally Modified Nanosized Silicate-Substituted Hydroxyapatite Co-Doped with Eu^3+^ and Sr^2+^ Ions

**DOI:** 10.3390/nano11010027

**Published:** 2020-12-24

**Authors:** Sara Targonska, Rafal J. Wiglusz

**Affiliations:** Institute of Low Temperature and Structure Research, Polish Academy of Sciences, Okolna 2, 50-422 Wroclaw, Poland

**Keywords:** spectroscopy, nanocrystallites, silicate-substituted hydroxyapatite, Eu^3+^ and Sr^2+^ ion co-doping, microwave-assisted hydrothermal method

## Abstract

In this paper, a series of structurally modified silicate-substituted apatite co-doped with Sr^2+^ and Eu^3+^ ions were synthesized by a microwave-assisted hydrothermal method. The concentration of Sr^2+^ ions was set at 2 mol% and Eu^3+^ ions were established in the range of 0.5–2 mol% in a molar ratio of calcium ion amount. The XRD (X-ray powder diffraction) technique and infrared (FT-IR) spectroscopy were used to characterize the obtained materials. The Kröger–Vink notation was used to explain the possible charge compensation mechanism. Moreover, the study of the spectroscopic properties (emission, emission excitation and emission kinetics) of the obtained materials as a function of optically active ions and annealing temperature was carried out. The luminescence behavior of Eu^3+^ ions in the apatite matrix was verified by the Judd–Ofelt (J-O) theory and discussed in detail. The temperature-dependent emission spectra were recorded for the representative materials. Furthermore, the International Commission on Illumination (CIE) chromaticity coordinates and correlated color temperature were determined by the obtained results.

## 1. Introduction

Nanotechnology has exerted a considerable impact on a vast number of scientific fields in the last decade. Studies on the preparation and characterization of apatite materials have been influenced by this. Nanoapatites are the focus of great research interest due to their biocompatible and nontoxic properties to encourage bone and tissue filling. The structure of the apatite allows for various modifications, providing the opportunity to create a material with intentional and targeted properties. Moreover, two unequal calcium positions can be substituted by ions with +1, +2 or +3 charges, such as Sr^2+^, Ba^2+^, K^+^, Na^+^, Mn^2+^, Li^+^, Mg^2+^ as well as lanthanide ions, etc. [1,2,3].

There is particular interest in the characterization of nanoapatite materials doped with optically active ions. Consequently, it is possible to successfully replace Ca^2+^ ion by Eu^3+^ [1,4,5], Ce^3+^ [6], Tb^3+^ [7], Dy^3+^ [8,9], Nd^3+^ [9,10], Sm^3+^ [8,9] ions to obtain materials with characteristic emission in red [1,4], green [11,12], violet [6] as well as blue [6] spectral regions. Recently, the apatite matrix has been the focus of attention and investigated in terms of white light emission. Promising materials are co-doped with Dy^3+^, Li^+^ and Eu^3+^ [13] or La^3+^, Dy^3+^ and Sr^3+^ ions [8]. The vast number of possibilities and favorable results encourage the detailed investigation of a variety of apatite modifications.

Apatite-based structures can tolerate numerous ionic substitutions in order to improve their properties for medical application. Recent research has shown that the combination of silica and strontium co-doping may improve their properties for medical application [14]. In vivo observations as well as in vitro studies have exposed the beneficial effects of using silica-based materials for bone treatment. It has been shown that silica promotes prolyl hydroxylase, stimulates the enzyme involved in collagen synthesis and participates in the proliferation and differentiation of bone mesenchymal stem cells and osteoblasts [15,16]. The synergy of great luminescence properties with bioactivity may result in obtaining a new group of specific materials dedicated to bioimaging and regeneration.

In this paper, we show the physicochemical characterization of silicate-substituted hydroxyapatite co-doped with Sr^2+^ and Eu^3+^ ions. Considerable attention has been paid to the luminescence properties, including emission and excitation spectra depending on Eu^3+^ ion concentration and heat-treating temperature as well as the influence of ambient temperature.

## 2. Materials and Methods

### 2.1. Synthesis of the Co-Doped Materials

Synthesizing of silicate-substituted hydroxyapatite co-doped with Eu^3+^ and Sr^2+^ ions involved a hydrothermal process. As substrates, the following were used: Ca(NO_3_)_2_∙4H_2_O (99.0–103.0% Alfa Aesar, Haverhill, MA, USA), (NH_4_)_2_HPO_4_ (>99.0% Acros Organics, Schwerte, Germany), Eu_2_O_3_ (99.99% Alfa Aesar, Haverhill, MA, USA), Sr(NO_3_)_2_ (99.0% min Alfa Aesar, Haverhill, MA, USA) and tetraethyl orthosilicate TEOS (>99% Alfa Aesar, Haverhill, MA, USA). The concentration of strontium ions for all obtained materials was fixed to 2 mol% in a ratio of calcium ion molar content. Moreover, the concentration of optical active Eu^3+^ ions was set to 0.5; 1.0 and 2.0 mol% in a ratio to the calcium ion molar content. A stoichiometric number of substrates were dissolved separately in deionization water (see Appendix A). A stoichiometric amount of Eu_2_O_3_ was digested in an excess of HNO_3_ (65% suprapure Merck KGaA, Darmstadt, Germany) to generate water-soluble Eu(NO_3_)_3_·xH_2_O. Afterwards, all starting substrates were added and mixed into a Teflon vessel. The ammonia solution (NH_3_∙H_2_O 25% Avantor, Poland) was used to obtain a pH level of around 10. The hydrothermal process was conducted in a microwave reactor (ERTEC MV 02-02, Wrocław, Poland) for 90 min at elevated temperature (250 °C) and under autogenous pressure (42–45 bar). The achieved materials were centrifuged, cleaned by deionization water several times and dried for 24 h. Then, the obtained materials were heat-treated in the temperature range of 400–600 °C for 3 h, increased in steps of 3.3 °C/min.

### 2.2. Physical–Chemical Characterization

X-ray powder diffraction studies were carried out using a PANalytical X’Pert Pro X-ray diffractometer (Malvern Panalytical Ltd., Malvern, UK) equipped with Ni-filtered Cu *Kα*_1_ radiation (*Kα*_1_ = 1.54060 Å, *U* = 40 kV, *I* = 30 mA) in the *2*θ range of 10°–70°. XRD patterns were analyzed by Match! software version 3.7.0.124.

The surface morphology and the element mapping were assessed by a FEI Nova NanoSEM 230 scanning electron microscope (SEM, Hillsboro, OR, USA) equipped with EDS spectrometer (EDAX GenesisXM4) and operating at an acceleration voltage in the range 3.0–15.0 kV and spots at 2.5–3.0 were observed. EDX analysis was carried out to confirm the chemical formula.

The Thermo Scientific Nicolet iS50 FT-IR spectrometer (Waltham, MA, USA) equipped with an Automated Beamsplitter exchange system (iS50 ABX containing DLaTGS KBr detector), built-in all-reflective diamond ATR module (iS50 ATR), Thermo Scientific Polaris™, was used to record the Fourier-transformed infrared spectra. As an infrared radiation source, we used the HeNe laser. FT-IR spectra of the powders were recorded in KBr pellets at 295 K temperature in the middle infrared range, from 4000 to 500 cm^−1^, with a spectral resolution of 2 cm^−1^.

### 2.3. Spectroscopy Properties

The emission, excitation emission spectra and luminescence kinetics were recorded by an FLS980 Fluorescence Spectrometer (Edinburgh Instruments, Kirkton Campus, UK) from Edinburgh Instruments equipped with a 450 W Xenon lamp and a Hamamatsu R928P photomultiplier. The hydroxyapatite powders were placed into a quartz tube. The excitation of 300 mm focal length monochromator was in Czerny–Turner configuration. All spectra were corrected during measurement according to the characteristics of the intensity of the excitation source. All spectra were recorded at room temperature. The spectral resolution of the excitation and emission spectra was 0.1 nm. Excitation spectra were recorded, monitoring the maximum of the emission at 618 nm that relates to the ^5^D_0_ → ^7^F_2_ transition, and emission spectra were recorded upon excitation wavelength at 394 nm. Before analysis, the emission spectra were normalized to the ^5^D_0_ → ^7^F_1_ magnetic transition. The luminescence kinetics profiles were recorded according to the ^5^D_0_ → ^7^F_2_ electric dipole transition.

Temperature-dependent emission spectra were recorded using the laser diode (*λ_exc_* = 375 nm), and as an optical detector, we used the Hamamatsu PMA-12 photonic multichannel analyzer (Hamamatsu Photonics K.K., Hamamatsu City, Japan). The presented emission spectra are the average result of 15 measurements with an exposure time of 500 ms.

## 3. Results and Discussion

### 3.1. X-ray Diffraction

The formation of the silicate-substituted hydroxyapatite crystalline nanopowders was investigated by the XRD measurements and is shown in Figure 1 as a function of the concentration of optically active ions as well as heat-treating process. All samples prepared via the hydrothermal method have shown detectable crystallinity for the entire range of sintering temperatures (400–600 °C for 3 h). The presence of the single phase of the final products was confirmed by the reference standard of hexagonal strontium-substituted hydroxyapatite ICSD-75518 [17]. No other phase was detected in the studied powders, indicating that dopant ions were completely dissolved in the silicate-substituted host lattice.

Structural refinement was carried out using the Maud software version 2.93 [18,19] and was based on apatite crystals with a hexagonal structure using better approximations, as well as indexing of the crystallographic information file (CIF). The formation of the hexagonal phase, as well as the successful incorporation of Eu^3+^ and Sr^2+^ ions into the apatite lattice, was confirmed by the results. The average grain sizes of silicate-substituted hydroxyapatite nanopowders were in the range of 16 to 56 nm (see Appendix A). The representative SEM image is presented in Appendix A. The effect of dopant ion substitution is confirmed by EDS measurements (see Appendix A).

The most intense diffraction peaks corresponding to hydroxyapatite structures are located at 25.9° (002), 31.7° (211), 32.2° (112), 32.9° (300) and 34.0° (202), assigned to the crystallographic planes in brackets. In the obtained materials, Eu^3+^ and Sr^2+^ ions replaced Ca^2+^ ions. In the apatite host lattice, Ca^2+^ ions are located in two different sites with various chemical and structural environments, Ca(1) and Ca(2) sites with C_3_ and C_S_ symmetry, respectively. The Ca(1) site is surrounded by nine oxygen atoms coming from PO_4_^3−^ groups, which formed a tricapped trigonal prism with formula CaO_9_. The Ca(2) site is an irregular polyhedron with formula CaO_6_OH formed by six oxygen atoms from PO_4_^3−^ and one hydroxyl group [3,20]. The difference between the ionic radii of trivalent europium and divalent strontium ions permit the occupation of two possible crystallographic positions of Ca^2+^ ions in the apatite host lattice (Ca^2+^ (CN_9_) = 1.18 Å, Eu^3+^ (CN_9_) = 1.12 Å, Sr^2+^ (CN_9_)−1.31 Å (Ca(1) site); Ca^2+^ (CN_7_) = 1.06 Å and Eu^3+^ (CN_7_) = 1.01 Å, Sr^2+^ (CN_7_)−1.21 Å (Ca(2) site)) [5,21].

### 3.2. Kröger–Vink Notation

The cationic vacancies formed by replacing Ca^2+^ ions with Eu^3+^ ions with higher charge could be balanced by SiO_4_^4−^ ions substituted into the PO_4_^3−^ position in the silicate-substituted apatite matrix. The occupation of divalent calcium ion sites by trivalent europium ions could be described according to the Kröger–Vink notation by the charge compensation phenomenon. According to this theory, the total charge in the material should be compensated by the creation of relatively positive or negative charge. The following processes may be observed:

A double negative vacancy on the Ca^2+^ position (V″_Ca_) is created by the substitution of divalent calcium ions by trivalent europium ions (Equation (1)):Eu_2_O_3_ + 3Ca^*^_Ca_ → 2Eu·_Ca_ + 3CaO + V″_Ca_(1)

The substitution of divalent calcium ions by trivalent rare earth ions could be explained by the creation of interstitial oxygen O^″^_i_ with double relative negative charge. The mechanism could be described as follows (Equation (2)):Eu_2_O_3_ + 2Ca^*^_Ca_ → 2Eu∙_Ca_ + 2CaO + O^″^_i_(2)

Eu^3+^ ions first replace into the Ca(1) site and this preference changes with increasing Eu^3+^ concentration in favor of the Ca(2) site. In case of substitution into the Ca(2) site, where calcium(II) ions are surrounded with one hydroxyl group and six oxygen atoms from PO_4_^3−^, these hydroxyl groups could participate in the charge compensation mechanism, expressed as (Equation (3)):Eu_2_O_3_ + 2Ca^*^_Ca(2)_ + 2OH^*^_OH_ → 2Eu∙_Ca(2)_ + 2O^′^_OH_ + 2CaO + H_2_O(3)

In the case of the obtained materials, the substitution of the PO_4_^3−^ group by the more negative SiO_4_^4−^ group could create a negative charge on the PO_4_^3−^ position and two positive charge vacancies (V^−^_Ca_) on the hydroxyl group position. The mechanism can be expressed as follows (Equation (4)):2SiO_2_ + 2PO_4_^*^_PO4_ + 2OH^*^_OH_ → 2SiO_4′PO4_ + 2V∙_OH_ + P_2_O_5_ + H_2_O(4)

In the present study, the charge compensation mechanism could be described as a combination of Equations (1)–(4). Equation (5) combines the creation of negative and positive vacancy, because of Ca^2+^ substitution by Eu^3+^ and PO_4_^3−^ substitution by SiO_4_^4−^, respectively.
2SiO_2_ + Eu_2_O_3_ + 2Ca^*^_Ca_ + 2PO_4_^*^_PO4_ → 2Eu∙_Ca_ + 2SiO_4′PO4_ + 2 CaO + P_2_O_5_(5)

### 3.3. Infrared Spectra

To confirm the presence of phosphate, silicate and hydroxyl groups, the infrared spectra were measured and are presented in Figure 2. According to previous reports, characteristic peaks are ascribed to the compound of hydroxyapatite [3,22,23]. The triply degenerated antisymmetric stretching vibration ν_3_(PO_4_^3−^) of phosphate groups is observed at 1101.1 and 1049.1 cm^−1^. At 966.4 cm^−1^, lines are detected which can be described as non-degenerated symmetric stretching bands ν_1_(PO_4_^3−^) vibrations. Strong absorption bands associated with the ν_4_(PO_4_^3−^) triply degenerated vibrations are located at 566.2 and 604.8 cm^−1^. Two bands related to the stretching and bending modes of OH^−^ groups are observed at 3571.3 and at 604.5 cm^−1^, respectively. These bands clearly confirm the presence of hydroxyl groups in the crystal structure. The broad peak between 3600 and 3200 cm^−1^ belongs to H_2_O vibration. Peaks assigned to (SiO_4_)^4−^ vibrational modes and Si–O–Si stretch modes are observed at 890 and 478 cm^−1^, respectively. It should be pointed out that there are similarly located vibrational modes of the (PO_4_)^3−^ and the silicate groups in the hydroxyapatite matrix, causing some interpretation problems. The Si–O symmetric stretching mode is located at 945 cm^−1^ and the weak peak corresponding to the P–O symmetric stretching mode is located at 962 cm^−1^ [23].

### 3.4. Spectroscopy Properties

The emission excitation spectra of the silicate-substituted appetites were recorded as a function of the europium ion concentration and heat-treated temperature in Figure 3a,b, respectively. The spectra were recorded at 300 K at an observation wavelength of 616 nm (16,233 cm^−1^). The presented spectra were normalized to the most intensity bands. In relation to the most intense band of the ^5^D_0_ → ^7^F_2_ transition, the spectra were recorded at an observation wavelength of 618 nm. In the UV range was observed a broad, intense band ascribed to the O^2−^ → Eu^3+^ charge transfer (CT) transition with a maximum located around 205 nm (48,780 cm^−1^). Increasing the dopant concentration and heat-treating temperature does not have an influence on the CT maximum position. In the composition of the excitation spectra were recorded sharp, narrow bands, attributed to the 4f–4f transitions of Eu^3+^ ions, at: ^7^F_0_ → ^5^F_(4,1,3,2)_, ^3^P_0_ at 299.3 nm (33,411 cm^−1^), ^7^F_0_ → ^5^H_(6,5,4,7,3)_ at 319.8 nm (31,269 cm^−1^), ^7^F_0_ → ^5^D_4_, ^5^L_8_ at 363.7 nm (27,495 cm^−1^), ^7^F_0_ → ^5^G_2_, ^5^L_7_, ^5^G_3_ at 383.7 nm (26,062 cm^−1^), ^7^F_0_ → ^5^L_6_ at 394.4 nm (25,354 cm^−1^), ^7^F_0_ → ^5^D_3_ at 413.8 nm (24,166 cm^−1^) and ^7^F_0_ → ^5^D_2_ at 465.1 nm (21,500 cm^−1^). In lanthanide ions, strongly isolating the 4f orbitals by the external 5s, 5p and 5d shells causes only slight changes in the positions of the electronic transition bands.

Figure 4a represents the emission spectra for the 2 mol% Eu^3+^-doped sample as a function of sintering temperature. Figure 4b shows the emission spectra as a function of optically active ion concentration for samples sintered at 600 °C. The emission spectra were detected at an excitation wavelength of 394 nm to directly excite the f electrons of Eu^3+^ ions. All spectra were normalized according to the ^5^D_0_ → ^7^F_1_ magnetic transition. As should have been expected, the emission transitions of the Eu^3+^ ions are forbidden by selection rules but do not consider the subtle influence of atom vibrations which consequently change the dipole moment, causing the occurrence of forbidden transitions on the spectrum. In the spectra are presented emission lines due to the ^5^D_0_ → ^7^F_J_ transitions for J = 0, 1, 2, 3 and 4, typical of Eu^3+^ ions, which are ascribed in reference to previous reports [5,24,25,26,27]. The bands located at the listed wavelength were attributed to the following transition: ^5^D_0_ → ^7^F_0_ at 577 nm (17,331 cm^−1^); ^5^D_0_ → ^7^F_1_ at 588 nm (17,006 cm^−1^); ^5^D_0_ → ^7^F_2_ at 616 nm (16,233 cm^−1^); ^5^D_0_ → ^7^F_3_ at 653 nm (15,313 cm^−1^) and ^5^D_0_ → ^7^F_0_ at 700 nm (14,285 cm^−1^). In general, the spectroscopic properties can provide information about the local chemical environment of Eu^3+^ ions. The essential change in emission spectra can be dependent on the quantity of possible crystallographic positions and therefore the amount of potential sites of substitution, as well as the presence of defects, additional phases or impurities [3,27].

The typical emission spectrum of 4f–4f electrons of Eu^3+^ ions is recorded in the red range of the electromagnetic radiation spectrum. In this range, the ^5^D_0_ → ^7^F_0,1,2_ transitions are observed and the shape of the lines in this region is correlated with the structural properties of the material. Due to these transitions, the trivalent europium ion is called an optical probe. If the Eu^3+^ ion is located in a centrosymmetric crystal lattice, the ^5^D_0_ → ^7^F_0_ transition is not observed. In the contrary situation, if the Eu^3+^ ion is placed in a non-centrosymmetric lattice, the ^5^D_0_ → ^7^F_0_ transition is detected on the emission spectra. Moreover, this is possible only in the low-symmetry crystal position as in C_n_, C_nv_ as well as C_S_ symmetry. In the hydroxyapatite matrix, calcium ions are located at two different crystal positions: Ca(1) and Ca(2) with local symmetry at C_3_ and C_S_, respectively. Both calcium positions could be occupied by Eu^3+^ ions. The additional crystallographic position appears as a result of reverse cis and trans symmetry of the Ca(2) site with the same point symmetry. In the environment of the Ca(1) site, nine oxygen atoms from phosphate groups are present. The calcium ion located on the Ca(2) site is in sevenfold coordination with six oxygen atoms from the phosphate group and one from the hydroxyl group [28]. The number and ratio of the intensity of the lines in the range between 570 and 580 nm give information about Eu^3+^ ion site-occupied preference. The coordination polyhedra of the Ca(1) and Ca(2) cations are presented in Figure 5a,b, respectively.

Emission spectra recorded for the investigated samples in the range of 570 to 580 nm are presented in Figure 5. In the emission spectra of 2 mol% Sr^2+^ and 0.5 mol% Eu^3+^ co-doped silicate-substituted hydroxyapatites, the transition attributed to ^5^D_0_ → ^7^F_0_ presents some interesting features due to abnormally strong intensity. This fact can be related to an important perturbation of the symmetry of the dopant induced by the introduction of silicate groups. Moreover, the abnormally strong intensity of the ^5^D_0_ → ^7^F_0_ transition has been reported in apatites such as oxyapatite Ca_10_(PO_4_)_6_O_2_, fluoroapatite Ca_5_(PO_4_)_3_F, hydroxyapatite Sr_10_(PO_4_)_6_(OH)_2_, or silicophosphate apatite Sr_5_(PO4)_2_SiO_4_, etc. This intense emission is attributed to the existence of the strong covalence of the Eu^3+^-O^2−^ bond in the Ca(2) site in the apatite lattice [28,29,30].

In the other cases, the population of the Ca^2+^ position replaced by the Eu^3+^ ion could be related to the thermal diffusion process of dopant ions in the apatite structure. It is commonly known that in the case of as-prepared apatite materials, only the emission associated with one type of site with C_3_ symmetry was observed, whereas with an increase in the calcination temperature, additional 0–0 peaks appeared [1,31,32]. The emission intensity ratio Ca(1)/Ca(2) has been calculated and results are presented in Table 1, showing values from 2.4 for as-prepared samples to 3.1 for sintered samples at 600 °C. Taking into account all results, one can note that in the silicate-substituted strontium-doped hydroxyapatite matrix, the Ca(1) site is more occupied by Eu^3+^ ions.

To obtain additional insight into the luminescence behavior of Eu^3+^ ions in silicate-substituted strontium co-doped apatite, the Judd–Ofelt theory was applied [33,34]. The results of the calculation of radiative (A_rad_), non-radiative (A_nrad_) and total (A_tot_) processes, as well as intensity parameters (Ω_2_, Ω_4_), quantum efficiency (η) and asymmetry ratio (R), are presented in Table 1. These parameters were calculated based on emission spectra and decay profiles according to equations defined in previous reports [35,36].

A comparison of the Judd–Ofelt intensity parameters (Ω_2_ and Ω_4_) is made between the samples with different Eu^3+^ ion concentrations and sintering temperatures. An increase in the Ω_2_ value is noted with an increase in the heat-treating temperature. This result indicates the increasingly hypersensitive character of the ^5^D_0_ → ^7^F_2_ transition and the increasing polarization of the Eu^3+^ ion environment. Consequently, Eu^3+^ ions in the material heat-treated at the highest temperature are in a more polarizable environment. The influence of silicon group presence is analyzed in comparison with previous work by our group [37]. A decrease in the J-O Ω_2_ parameter is noted from 6.683 × 10^−20^ cm^2^ (see [37]) to 4.506×10^−20^ cm^2^ (see Table 1) for Ca_9.7_Sr_0.2_Eu_0.1_(PO_4_)_6_(OH)_2_ and for Ca_9.7_Sr_0.2_Eu_0.1_(PO_4_)_2_(SiO_4_)_4_(OH)_2_, respectively. This observation can be related to the improvement of the Eu^3+^ cation polyhedral and to the decrease in the covalence character of the Eu^3+^–O^2−^ bond in silicate-substituted apatite. In the case of the investigated samples, the Ω_2_ > Ω_4_ parameter suggests that Eu^3+^ ions are not located in the local symmetry of centrosymmetric character. The calculated results are in agreement with previous reports regarding apatite systems [1,24,35,37].

The highest quantum efficient (η) is observed for the 1 mol% Eu^3+^-doped compound with the value of 40%. The quantum efficient is reduced by the increase as well as the decrease of optically active ion concentration to 37% and 34%, respectively. Samples heat-treated at higher temperatures demonstrate higher quantum efficient values, and the η parameter increases by the maximum of 15%.

The luminescence intensity ratio (R) of the electric dipole transition ^5^D_0_ → ^7^F_2_ to the magnetic dipole transition ^5^D_0_ → ^7^F_1_ has been calculated and the results are presented in Table 1. As expected, the sintering process and concentration of optical active ions have a significant influence on the symmetry of the Eu^3+^ ion environment in the apatite matrix. The R factor increases with an increase in the heat-treating temperature, which indicates lower symmetry around Eu^3+^ ions and suggests that the covalence of Eu^3+^-O^2−^ is higher. The R parameter increases for 0.5 mol% Eu^3+^ to 1.0 mol% Eu^3+^ from 3.1 to 4.5, respectively. Then, the opposite trend is observed, and the R parameter decreases from 4.5 to 4.1 for 1.0 mol% Eu^3+^ and 2.0 mol% Eu^3+^.

Figure 6 presents the energy level diagram of Eu^3+^ corresponding to the detected excitation and emission spectra. As seen, Eu^3+^ ions were pumped to upper excited levels. The emission bands from the ^5^D_1_, ^5^D_2_ and ^5^D_3_ levels are not observed at room temperature in the case of the investigated samples, which suggests a fast, non-radiative (NR), multiphonon relaxation from the excited state ^5^L_6_ to the ^5^D_0_ state.

### 3.5. Temperature-Dependent Emission

To further study the possible application under high temperature, the temperature-dependent emission spectra of the silicate-substituted hydroxyapatite co-doped with 2 mol% Sr^2+^ and 2 mol% Eu^3+^ were measured and are presented in Figure 7. Emissions were recorded in the range of 80 to 725 K and at an excitation wavelength of 375 nm into the ^5^G_2_ level. It is seen that the emission intensity decreases clearly with an increase in the ambient temperature, but the decrease is not linear in the whole range of measured temperatures. For the most intense line corresponding to the ^5^D_0_ → ^7^F_2_ transition, the linear relationship applies between 80 and 325 K (R^2^ = 99.4%). Then, between 450 and 800 K, the decrease can be described by the exponential equation (see Appendix A). The line corresponding to the ^5^D_0_ → ^7^F_0_ transition of the Ca(2) calcium site (573 nm) is completely eliminated at 350 K.

In Figure 8 has been shown the Commission Internationale de l’Eclairage (CIE) 1931 chromaticity diagram for the Ca_9.6_Sr_0.2_Eu_0.2_(PO_4_)_2_(SiO_4_)_4_(OH)_2_ sample. The CIE color coordinates are listed in Appendix A, which are calculated from the temperature-dependent emission spectra [38]. It has been reported that the emission color changed from reddish-orange to orange with the increasing of ambient temperature, but the reddish-orange emission was stable until 750 K. These results show that the Eu^3+^-activated silicate-substituted apatites have the potential to be color-stable materials capable of operating at a wide range of ambient temperatures.

### 3.6. Decay Time

The luminescence kinetics corresponding to the ^5^D_0_ → ^7^F_2_ transition were obtained at room temperature. As expected, all the recorded decays presented non-single exponential character. This phenomenon was consistent with the existence of two non-equivalent Eu^3+^ positions and, because of this fact, the effective emission lifetime was calculated by Equation (6). The recorded decays and calculated luminescence lifetimes (τ) are presented in Figure 9, as a function of Eu^3+^ concentration (a) and (b) as well as heat-treating temperature ((c) and (d)), respectively.
(6)$$\tau_{m}=\frac{\int_{0}^{\infty }tI\left(t\right)dt}{\int_{0}^{\infty }I\left(t\right)dt}≅\frac{\int_{0}^{t_{max}}tI\left(t\right)dt}{\int_{0}^{t_{max}}I\left(t\right)dt}$$

## 4. Conclusions

In this study, it has been shown for the first time that a series of silicate-substituted hydroxyapatite co-doped with 2 mol% Sr^2+^ and Eu^3+^ ions in the range of 0.5–2.0 mol % in a ratio to the entire Ca^2+^ ion content were successfully synthesized by the hydrothermal method assisted with microwave and heat-treated. The average crystal sizes of the studied materials were in the range of 16–56 nm as calculated by the Rietveld method.

Attention was paid to the structural and spectroscopic properties related to a variable amount of Eu^3+^ ion concentration. The spectroscopic properties have shown for the sintered samples that Eu^3+^ ions occupied three independent crystallographic sites: one Ca(1) site with *C*_3_ local symmetry and two Ca(2) sites with *C_s_* local symmetry with *cis* and *trans* symmetry. The ^5^D_0_ → ^7^F_2_ hypersensitive transition is the most intense for most obtained materials, excluding the sample co-doped with 0.5 mol% Eu^3+^ ion and sintered at 600 °C, where the most dominant is ^5^D_0_ → ^7^F_0_ transition. Moreover, the charge compensation mechanism in the materials induced by the substitution of Eu^3+^ and Sr^2+^ ions into the silicate-substituted hydroxyapatite host lattice was rendered in the Kröger–Vink notation.

The luminescence decay times corresponding to the most intense ^5^D_0_ → ^7^F_2_ transition were recorded. The luminescence kinetics was characterized by a non-exponential decay profile and was in the range of 2.15 ms (0.5 mol% Eu^3+^) and 19.4 ms (1 mol% Eu^3+^) to 1.92 ms (2 mol% Eu^3+^) for the samples sintered at 600 °C. On the other hand, the decay times for the samples doped with 2 mol% Eu^3+^ as a function of sintered temperature were in the range of 15.2 to 1.92 ms. The typical modes of the vibrations of the silicate-substituted hydroxyapatite ion group were detected in the FT-IR spectra, and these included the OH^−^ group vibrations characteristic of the hydroxyapatite matrix. The simplified Judd–Ofelt theory was used for a detailed analysis of the luminescence spectra. The hydroxyapatite containing 1 mol% of Eu^3+^ ions was evaluated to be the most optically efficient material among all the studied silicate-substituted hydroxyapatites. The International Commission on Illumination (CIE) color coordinates showed that the emission color can be tuned by varying the ambient temperature. The emission color was changed from reddish-orange to orange.

## Figures and Tables

**Figure 1 nanomaterials-11-00027-f001:**
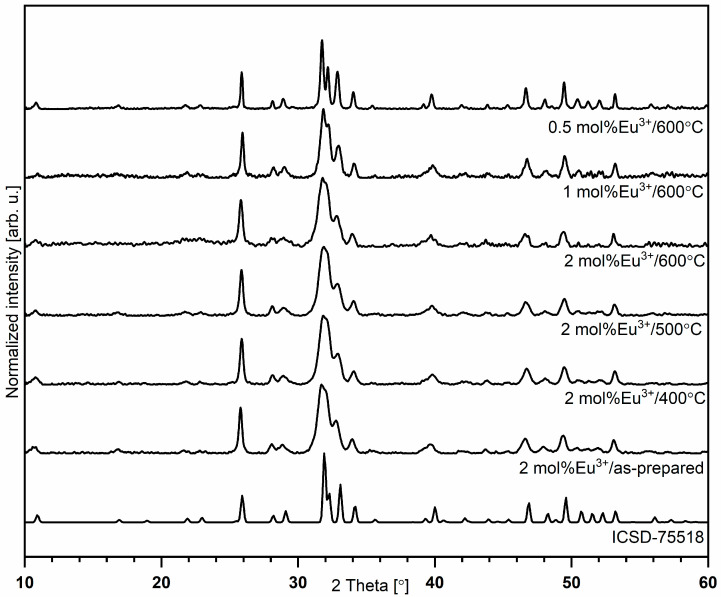
X-ray diffraction pattern of silicate-substituted hydroxyapatite co-doped with 2 mol% Sr^2+^ and x mol% Eu^3+^ ions.

**Figure 2 nanomaterials-11-00027-f002:**
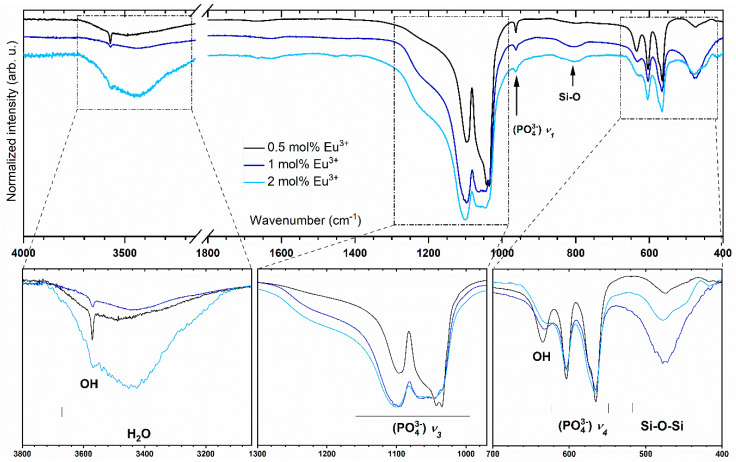
FT-IR spectra of the silicate-substituted hydroxyapatite co-doped with Sr^2+^ (2.0 mol%) and Eu^3+^ (0.5, 1.0 and 2.0 mol%) ions.

**Figure 3 nanomaterials-11-00027-f003:**
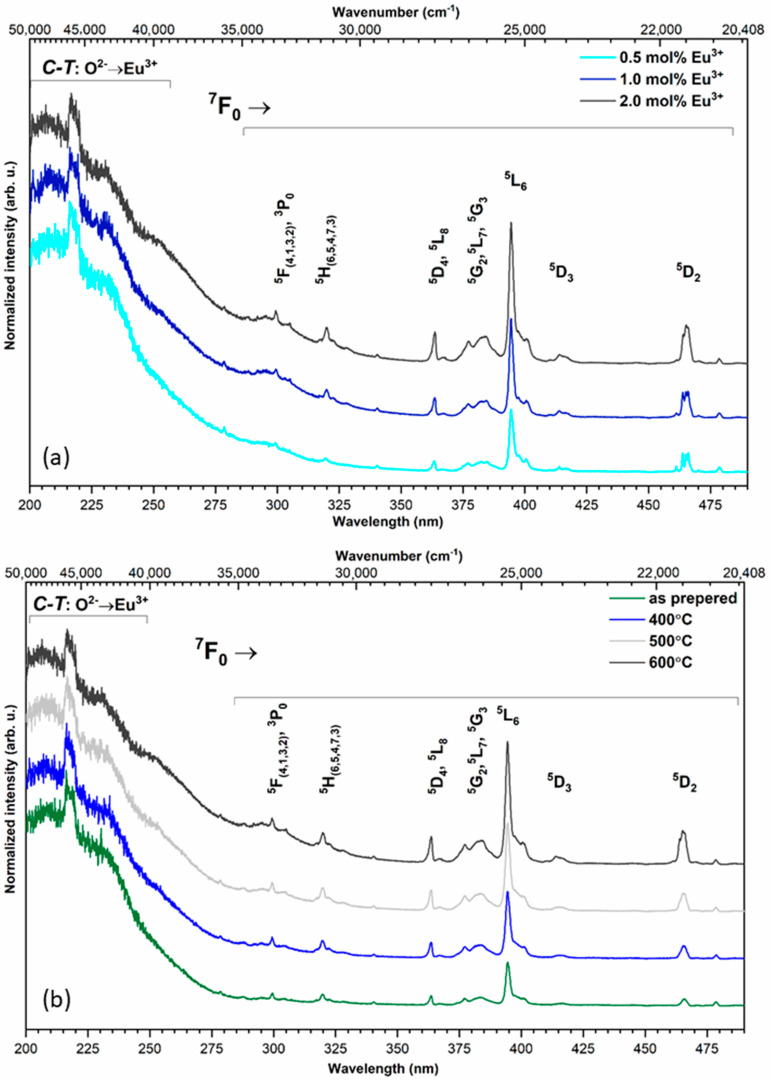
Excitation emission spectra of Ca_9.8-x_Sr_0.2_Eu_x_(PO_4_)_2_(SiO_4_)_4_(OH)_2_, where x = 0.5, 1.0, 2.0 mol%, sintered at 600 °C (**a**) as well as Ca_9.6_Sr_0.2_Eu_0.2_(PO_4_)_2_(SiO_4_)_4_(OH)_2_ as a function of sintering temperature (**b**).

**Figure 4 nanomaterials-11-00027-f004:**
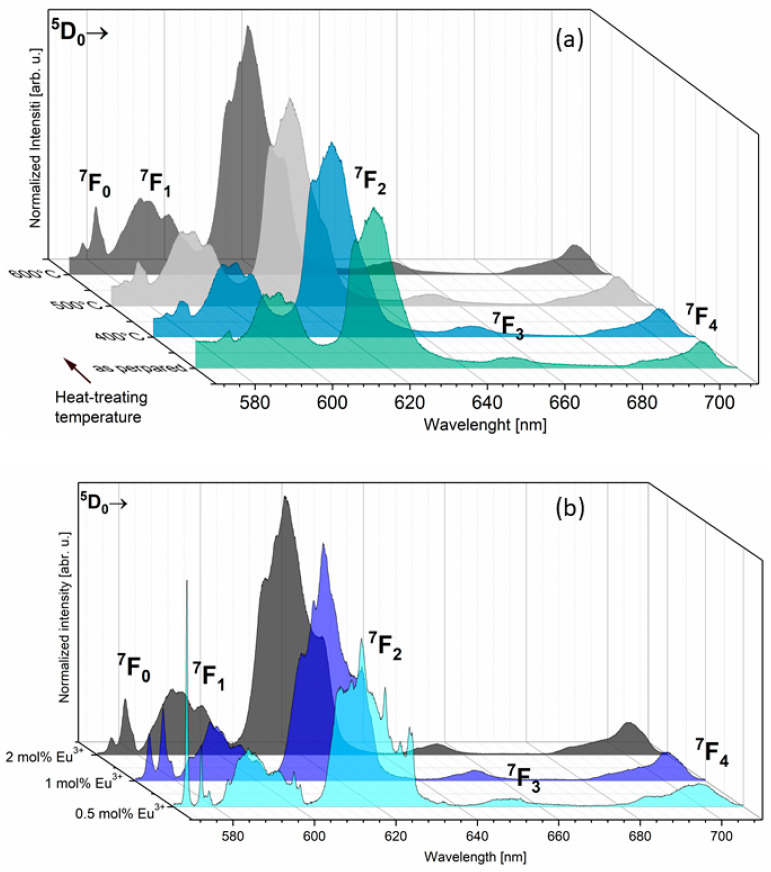
Emission spectra of Ca_9.6_Sr_0.2_Eu_0.2_(PO_4_)_2_(SiO_4_)_4_(OH)_2_ as a function of sintering temperature (**a**) and Ca_9.8-x_Sr_0.2_Eu_x_(PO_4_)_2_(SiO_4_)_4_(OH)_2_, where x = 0.5, 1.0, 2.0 mol%, sintered at 600 °C (**b**).

**Figure 5 nanomaterials-11-00027-f005:**
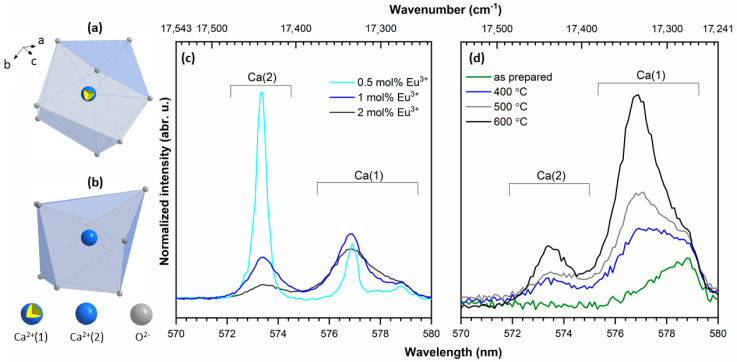
The projection of the coordination polyhedra of (**a**) Ca(1) and (**b**) Ca(2) cations, respectively. Emission spectra of (**c**) Ca_9.8-x_Sr_0.2_Eu_x_(PO_4_)_2_(SiO_4_)_4_(OH)_2_, where x = 0.5, 1.0, 2.0 mol%, sintered at 600 °C and (**d**) Ca_9.6_Sr_0.2_Eu_0.2_(PO_4_)_2_(SiO_4_)_4_(OH)_2_ as a function of sintering temperature, for the ^5^D_0_ → ^7^F_0_ transition.

**Figure 6 nanomaterials-11-00027-f006:**
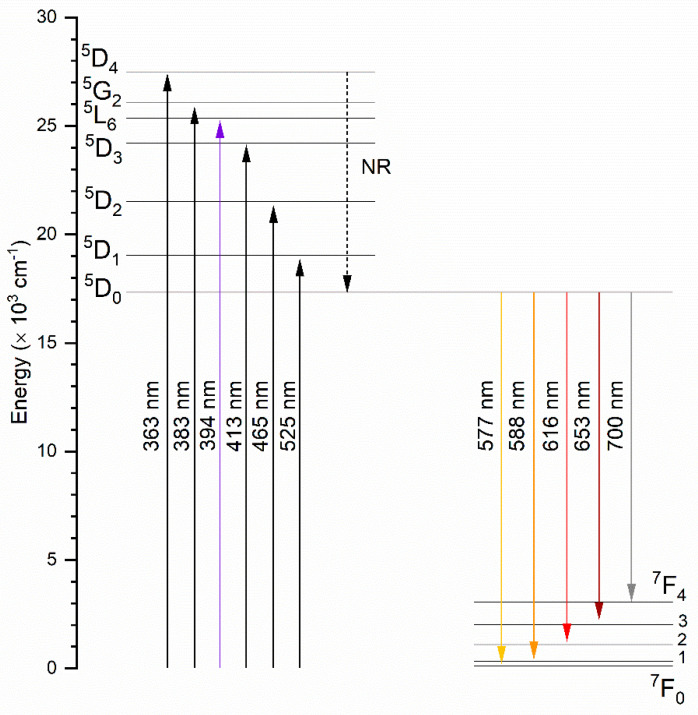
The simplified energy level scheme for Eu^3+^ ion in silicate-substituted strontium-doped hydroxyapatite.

**Figure 7 nanomaterials-11-00027-f007:**
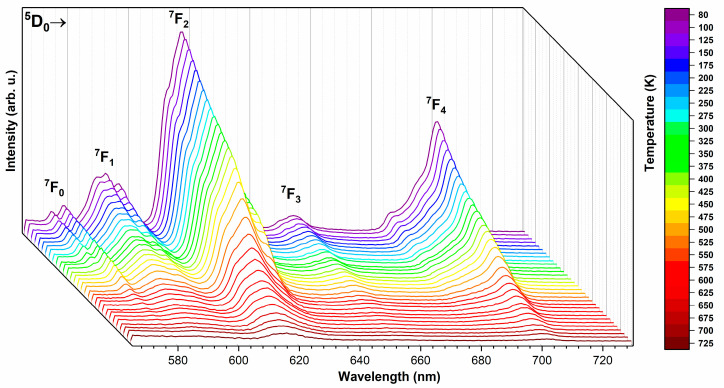
Temperature-dependent emission spectra of the Ca_9.6_Sr_0.2_Eu_0.2_(PO_4_)_2_(SiO_4_)_4_(OH)_2_, sintered at 600 °C.

**Figure 8 nanomaterials-11-00027-f008:**
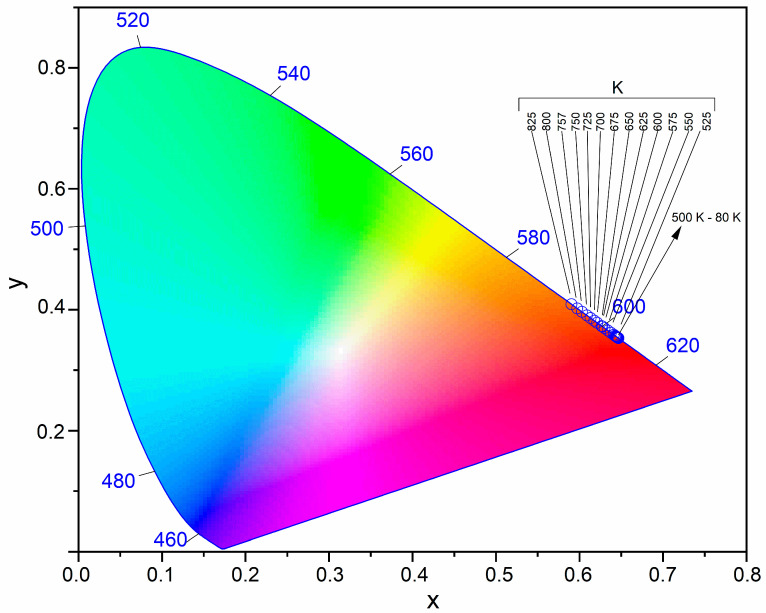
CIE 1931 chromaticity diagram of the Ca_9.6_Sr_0.2_Eu_0.2_(PO_4_)_2_(SiO_4_)_4_(OH)_2_ as a function of ambient temperature.

**Figure 9 nanomaterials-11-00027-f009:**
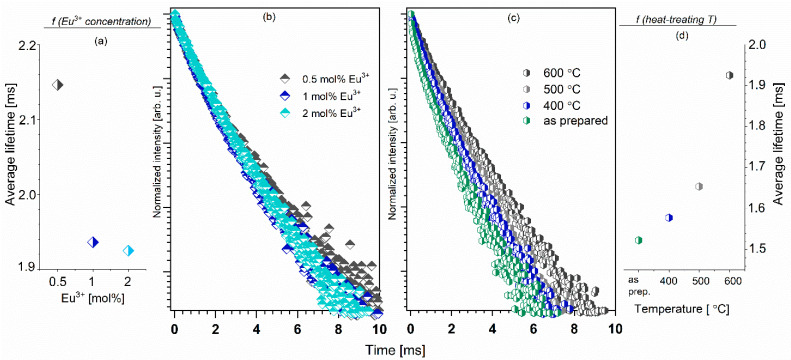
Calculated average lifetimes (**a**) as well as luminescence decay profiles (**b**) of Ca_9.8-x_Sr_0.2_Eu_x_(PO_4_)_2_(SiO_4_)_4_(OH)_2_, sintered at 600 °C, as a function of Eu^3+^ ion concentration. Luminescence decay profiles (**c**) and calculated average lifetimes (**d**) of the Ca_9.6_Sr_0.2_Eu_0.2_(PO_4_)_2_(SiO_4_)_4_(OH)_2_ as a function of sintering temperature.

**Table 1 nanomaterials-11-00027-t001:** Decay rates of radiative (A_rad_), non-radiative (A_nrad_) and total (A_tot_) processes of ^5^D_0_ → ^7^F_J_ transitions, luminescence lifetimes (τ), intensity parameters (Ω_2_, Ω_4_), quantum efficiency (η) and asymmetry ratio (R) for investigated samples.

Sample	Ca_9.8-x_Sr_0.2_Eu_x_(PO_4_)_2_(SiO_4_)_4_(OH)_2_ 600 °C			Ca_9.6_Sr_0.2_Eu_0.2_(PO_4_)_2_(SiO_4_)_4_(OH)_2_			
	x = 0.5 mol% Eu^3+^	x = 1.0 mol% Eu^3+^	x = 2.0 mol% Eu^3+^	as prepared	400 °C	500 °C	600 °C
Ca(1)/Ca(2)	0.42	2.08	3.06	2.46	2.64	2.78	3.06
A_rad_ (s^−1^)	159.02	207.91	192.34	147.11	158.88	168.89	192.34
A_nrad_ (s^−1^)	306.85	308.56	327.17	509.69	475.04	435.80	327.17
A_tot_ (s^−1^)	465.87	516.48	519.51	656.80	633.92	604.69	519.51
τ (ms)	2.15	1.94	1.92	1.52	1.58	1.65	1.92
Ω_2_ (10^−20^ cm^2^)	4.084	5.898	5.377	3.678	4.166	4.506	5.377
Ω_4_ (10^−20^ cm^2^)	1.078	1.119	0.994	0.997	0.907	0.977	0.995
h (%)	34.13	40.26	37.02	22.40	25.06	27.93	37.02
R	2.85	4.11	3.75	2.57	2.91	3.14	3.75

## Data Availability

Not applicable.

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
