# Peer review of "Investigation of Physicochemical Properties of the Structurally Modified Nanosized Silicate-Substituted Hydroxyapatite Co-Doped with Eu3+ and Sr2+ Ions"

_nanomaterials, 2020, doi:10.3390/nano11010027_

Round 1
Reviewer 1 Report
- The materials and methods section is too brief and does not provide an option to reproduce experiments. More detailed information with using grams and milliliters, as well as the solution concentrations are required.
- Please, re-check all formulae and ions. There are cases, when super- or sub-scripts are not used.
- Chemical formula Ca10(PO4)2(SiO4)4(OH)2 appears to be charge unbalanced. Ca is +2, thus 10Ca give +20. PO4 is -3, thus (PO4)2 give -6. SO4 is -4, thus (SO4)4 give -16. In addition, there are (OH)2 which give -2. Thus, either 1 more Ca must be added or (OH)2 must be omitted. Please, based on your results, select the correct option and PROVIDE EXPERIMENTAL EVIDENCES for your selection. Similar is valid for Sr0.2Eu0.2Ca9.6(PO4)4(SiO4)2(OH)2.
- Fig. 2: SEM Image is senseless (there are many thousands of fine materials, which look as that on the photo). Either provide the specific image or move it to the supplementary.
- P2O5 is mentioned in eq. (4) and (5). Did you really detect its formation?
- Line 182: “vibrational modes of the (PO3)4- and …” – perhaps, (PO4)3-?
- Alas, my background and knowledge do not allow me to evaluate sections 3.4 - 3.6 and Figs. 4-10. Another review by a spectroscopic (emission, emission excitation and emission kinetics) professional appears to be necessary.
Author Response
Dear Editor,
We would like to express our sincerest gratitude to the Reviewers for their enormous efforts in criticizing the manuscript. We have considered all raised question here follows the detailed answers to the Reviewers. Moreover, all changes we have made to the original manuscript, are marked in the red colour in the text.
Review:
Comments and Suggestions for Authors
- The materials and methods section is too brief and does not provide an option to reproduce experiments. More detailed information with using grams and milliliters, as well as the solution concentrations are required.
Answer: We agree with Referee suggestion therefore, it has been added the detailed information about the substrates in Table S1 (see Supplementary information)
- Please, re-check all formulae and ions. There are cases, when super- or sub-scripts are not used.
Answer: The mistakes have been corrected and marked in the red colour.
- Chemical formula Ca10(PO4)2(SiO4)4(OH)2 appears to be charge unbalanced. Ca is +2, thus 10Ca give +20. PO4 is -3, thus (PO4)2 give -6. SO4 is -4, thus (SO4)4 give -16. In addition, there are (OH)2 which give -2. Thus, either 1 more Ca must be added or (OH)2 must be omitted. Please, based on your results, select the correct option and PROVIDE EXPERIMENTAL EVIDENCES for your selection. Similar is valid for Sr0.2Eu0.2Ca9.6(PO4)4(SiO4)2(OH)2.
Answer: The charge unbalance has been explained in the “3.2. Kröger-Vink-notation” Section. Due to this theory the total charge in the material should be compensated by creation of relatively positive or negative charge. The theory has been previously involved in the scientific literature [1–4].
- Fig. 2: SEM Image is senseless (there are many thousands of fine materials, which look as that on the photo). Either provide the specific image or move it to the supplementary.
Answer: The SEM image (Figure 2) has been moved to Supplementary Information File.
- P2O5 is mentioned in eq. (4) and (5). Did you really detect its formation?
Answer: The charge compensation phenomena explained by the Kröger-Vink-notation was used. The description P2O5 means the closer connection, which has not been changed. The charge compensation is done by created vacancy.
- Line 182: “vibrational modes of the (PO3)4- and …” – perhaps, (PO4)3-?
Answer: The corrections have been made in the text.
- Alas, my background and knowledge do not allow me to evaluate sections 3.4 - 3.6 and Figs. 4-10. Another review by a spectroscopic (emission, emission excitation and emission kinetics) professional appears to be necessary.
Answer: The authors have a thorough knowledge of the optical spectroscopy, especially about lanthanide ions.

Reviewer 2 Report
In this work, Targonska and Wiglusz investigated the physicochemical properties of nanosized silicate-substituted hydroxyapative co-doped with Eu3+ and Sr2+ ions.
This is a well-made study that should be of great interest for the readership of Nanomaterials. In fact, the characterization of the co-doped materials was made in detail and with good results. Given this, my recommendation is for acceptance after minor revision:
-What are the synthesis yields (wt. %) of the co-doped materials?
-What is the emission quantum yield of the co-doped materials?
-What is the excitation wavelength used to obtain the emission spectra presented in Figures 5, 6 and 8?
-The quality of used English is not very good;
-The authors should compare better their results with those of similar materials found in the literature;
Author Response
Dear Editor,
We would like to express our sincerest gratitude to the Reviewers for their enormous efforts in criticizing the manuscript. We have considered all raised question here follows the detailed answers to the Reviewers. Moreover, all changes we have made to the original manuscript, are marked in the red colour in the text.
Review:
Comments and Suggestions for Authors
In this work, Targonska and Wiglusz investigated the physicochemical properties of nanosized silicate-substituted hydroxyapative co-doped with Eu3+ and Sr2+ ions.
This is a well-made study that should be of great interest for the readership of Nanomaterials. In fact, the characterization of the co-doped materials was made in detail and with good results. Given this, my recommendation is for acceptance after minor revision:
-What are the synthesis yields (wt. %) of the co-doped materials?
Answer: The synthesis yields of the co-doped materials is approximately equal to 65 – 80 wt. %.
-What is the emission quantum yield of the co-doped materials?
Answer: The quantum yield was not measured for the studied materials. There are some reports showing the absolute emission quantum yield related to the Eu3+ ions doped hydroxyapatites. The values are in the range from 2 to 15 % [5,6].
-What is the excitation wavelength used to obtain the emission spectra presented in Figures 5, 6 and 8?
Answer: The presented emission spectra were obtained by using the λexc = 394 nm wavelength (see Figures 5 and 6) and the λexc = 304 nm wavelength (see Figure 8). This information has been contained in the manuscript.
-The quality of used English is not very good;
Answer: The English has been corrected.
-The authors should compare better their results with those of similar materials found in the literature;
Answer: In our opinion the presented results are compared with similar materials sufficiently. In the Section “3.1. X-ray Diffraction” presented measurements are compared with theoretical pattern, lines 109 – 110: “The presence of single phase of the final products was confirmed by the reference standard of hexagonal strontium substituted hydroxyapatite ICSD-75518 [17]”. The next Section “3.3. Infrared spectra” also includes reference to the literature, lines 169 – 170: “According to pervious papers characteristic peaks are ascribed to the compound of hydroxyapatite [3,22,23]”. In the “3.4. Spectroscopy properties” Section are included the references to our previous study as well as other researchers work with similar materials, lines 244 – 247: “the abnormal strong intensity of the 5D0 ® 7F0 transition has been reported in the apatites such as oxyapatite Ca10(PO4)6O2, fluoroapatite Ca5(PO4)3F, hydroxyapatite Sr10(PO4)6(OH)2, or silicophosphate apatite Sr5(PO4)2SiO4 etc. This intense emission is attributed to the existence of the strong covalence of the Eu3+-O2− bond in the Ca(2) site in apatite lattice [28–30]”; lines 249 – 252: “It is commonly known that in case of as-prepared apatite materials only the emission associated with one type of site with C3 symmetry was observed, whereas with an increase in the calcination temperature additional 0–0 peaks appeared [1,31,32]”; lines 271 – 272: “The influence of silicon group presence is analysed in comparison with previous work of our group [37]”; lines 277 – 278: “Calculated results are in agreement with previous papers regarding apatite systems [1,24,35,37]”.

Round 2
Reviewer 1 Report
The revised version looks better.